# Association Between Dreams, Anxiety, and Depressive Symptoms Among Japanese Adolescents: A Cross-Sectional Study

**DOI:** 10.3390/clockssleep7030034

**Published:** 2025-06-26

**Authors:** Yuki Tanaka, Yuichiro Otsuka, Suguru Nakajima, Osamu Itani, Tomomi Miyoshi, Yoshitaka Kaneita

**Affiliations:** 1Division of Public Health, Department of Social Medicine, Nihon University School of Medicine, Tokyo 173-8610, Japan; tanaka.yuki@nihon-u.ac.jp (Y.T.);; 2Department of Psychiatry, Nihon University School of Medicine, Tokyo 173-8610, Japan; 3Department of Public Health, International University of Health and Welfare, Narita 286-8686, Japan

**Keywords:** adolescence, anxiety, depression, dreams, cognitive rumination

## Abstract

Worsening adolescent mental health is a significant social issue. Although dreams may reflect one’s mental state, few studies have focused on adolescents. Therefore, this study investigated the relationship between dream content and mental health, specifically anxiety disorder and depressive symptoms, among Japanese adolescents. This cross-sectional study obtained data on gender, grade, age, lifestyle habits, weekday sleep duration, anxiety disorder symptoms, depressive symptoms, and dreams from Japanese high school students. The data were analyzed via multiple logistic regression analyses. The prevalence of anxiety and depressive symptoms increased with the frequency of “rumination at bedtime”, “memory of dreams”, “emotional carryover”, and “awakening by frightening”, “unpleasant”, “film-like”, “fantastical”, and “recurring” dreams. However, this was not the case for “pleasant dreams”. Furthermore, “rumination at bedtime” (anxiety disorder symptoms: adjusted odds ratio: 10.60; 95% confidence interval: 5.92–18.97; depressive symptoms: 8.79, 5.58–13.87) and “unpleasant dreams” (anxiety disorder symptoms: 5.25, 2.86–9.64; depressive symptoms: 10.13, 5.57–18.44) exhibited particularly high odds ratios. “Rumination at bedtime” and “unpleasant dreams” may serve as early indicators of declining mental health. School- and parent-led interventions aimed at improving mental well-being may help prevent the progression or exacerbation of anxiety and depressive symptoms among adolescents.

## 1. Introduction

According to the World Health Organization (WHO), approximately 4.4% of the global population (5.1% women and 3.6% men) experienced depressive symptoms, while 3.6% (4.6% women and 2.6% men) suffered from anxiety disorders in 2017 [1]. Over 60% of adults with mental illnesses develop these disorders during adolescence [2]. Furthermore, a 2021 United Nations Children’s Fund (UNICEF) report revealed that 14% of teenagers were diagnosed with a mental illness; in addition, suicide ranked as the fifth leading cause of death in this age group [3]. Consequently, anxiety disorders and depressive symptoms are recognized as significant public health concerns.

The recent worsening of mental health among Japanese children and students has emerged as a pressing social issue [4,5,6,7]. According to a cross-sectional survey conducted in 2022, 17.3% of high school students exhibited depressive symptoms (with scores of ≥ 10 on the Patient Health Questionnaire for Adolescents (PHQ-A)), while 19.0% exhibited generalized anxiety symptoms (with scores of ≥10 on the Generalized Anxiety Disorder-7 (GAD-7,)) [8]. Furthermore, surveys conducted between 2020 and 2023 revealed that 11.4% of children (from 5th to 11th grade) exhibited moderate-to-severe depressive symptoms in 2021, which rose to 13.3% in both 2022 and 2023 [9]. Among high school students, 16.3% and 34.3% exhibited moderate-to-severe and mild-to-severe depressive symptoms in 2023, respectively [9]. Moreover, suicide statistics for 2024 revealed a gradual increase among junior high and high school students, despite a decline in other age groups [10]. These findings highlight the deteriorating mental health among adolescents and urgent need for intervention.

Previous studies have identified various factors associated with mental health, such as stress, diet, exercise, trauma, relationships and social support, academics, gender, bullying victimization, and sleep quality [6,11,12,13,14,15,16]. In addition, dreams were also a relevant factor [17,18,19]. Notably, dreams serve as a salient indicator of one’s mental state [17,18,19]. Studies have revealed that daytime experiences and emotions influenced dream content [20]. Particularly, nightmares are linked to emotional states and may be associated with anxiety and depression [21,22,23]. Furthermore, individuals with anxiety disorders frequently experience fear-related dreams [24]. Emotional responses to dreams are influenced by individual differences in waking emotional tendencies and personality characteristics; hence, those who experience more positive or negative emotions while awake may reflect similar emotional patterns in their dreams [25].

Dreams that occur during rapid eye movement (REM) sleep are vivid and active [26,27]. Furthermore, REM sleep supposedly plays a vital role in processing daytime stress and anxiety and facilitating the consolidation of emotional memories [28,29]. Disruptions in REM sleep have been linked to impaired emotional processing and worsening depressive symptoms [30]. Furthermore, REM sleep deprivation increases anxiety, depression, and even hallucinations [31]. Nightmares resulting from failures in emotional regulation are associated with stress and trauma [32].

Studies have demonstrated a relationship between nightmares and anxiety disorder symptoms among adolescents, as well as the influence of emotions on dream content [22]. Furthermore, individual differences in dream recall, with emotionally intense dreams being more easily remembered, have also been highlighted [33]. Research suggests that dream content may reflect adolescents’ psychological health and dream analysis may serve as a valuable psychological assessment tool [34]. Furthermore, recurring dreams mirror the psychological state of early adolescents [35], while nightmares tend to decrease with age [36].

Hence, dreams play a significant role in both psychological and physiological health [32,35]. Additionally, they are a potential indicator of one’s mental and emotional state [21,24,37], as nightmares and emotionally intense dreams occur more frequently among individuals with anxiety and depression [28,36]. Analyzing dream content may aid in understanding and alleviating mental health conditions [34,35]. However, most studies have focused on adults, with limited focus on adolescents, and have been conducted in Western countries. Hence, cultural and regional differences remain largely unexplored. Additionally, whether dreams contribute to the worsening of one’s mental state or simply reflect it remains unclear.

Previous research has largely focused on nightmares and their associations with mental health. However, studies examining various dream characteristics, such as dream frequency, emotional tone, or thematic content, remain scarce, particularly among adolescents. These gaps highlight the need for further investigations into the role of dreams in adolescent mental health across different cultural contexts. Therefore, this study investigated how different dream experiences were related to mental health, specifically symptoms of anxiety disorders and depression, among Japanese high school students.

## 2. Results

### 2.1. Participants’ General Characteristics

Out of the 1060 students, we received responses from 863 students. Of these, nine students who did not fill in their gender or grade were excluded. Consequently, 854 valid participants remained (valid response rate: 80.6%) (Table 1).

### 2.2. Participants’ Characteristics and Prevalence of Anxiety Disorder and Depressive Symptoms Based on Participants’ Characteristics

Table 2 presents the participants’ characteristics and prevalence of anxiety disorder and depressive symptoms based on participants’ characteristics. The results revealed the number and percentage of participants in each category, along with the prevalence and *p*-values for depression and anxiety (*p* < 0.05 was considered statistically significant). Anxiety disorders were observed in 9.8% of men and 17.1% of women, while depressive symptoms were observed in 32.1% of men and 43.7% of women. Additionally, 46.0%, 40.5%, and 13.5% of the participants often, sometimes, and almost never experienced dreams, respectively. The presence of anxiety disorder symptoms differed significantly based on all the characteristics, except for “grade” and “exercise frequency”. Meanwhile, the presence of depressive symptoms differed significantly based on all the characteristics, except for “exercise frequency”.

### 2.3. Frequency of Dream-Related Experiences and Dream Content and Prevalence of Anxiety Disorder and Depressive Symptoms Based on the Frequency and Dream Content

Table 3 presents the frequency of each dream-related experience and dream content, as well as prevalence of anxiety disorder and depressive symptoms based on the frequency of these experiences and content. The results revealed the number and percentage of participants in each category of dream-related factors, along with the prevalence of depression and anxiety and their respective *p* for the trend values (with *p* < 0.05 considered statistically significant). For all dream-related experiences and types of dream content, the prevalence of anxiety disorder symptoms increased significantly with the rising frequency of these experiences and content, except for “pleasant dreams”. Similarly, for all dream-related experiences and types of dream content, the prevalence of depressive symptoms increased significantly with the increasing frequency of these experiences and content, except for “pleasant dreams”.

### 2.4. Association Between Frequency of Each Dream-Related Experience and Dream Content and Presence of Anxiety Disorder and Depressive Symptoms

Table 4 presents the relationship between the frequency of each dream characteristic and the presence of anxiety disorder and depressive symptoms. The results were presented as adjusted odds ratios (aOR) with 95% confidence intervals (CI). Statistical significance was defined as *p* < 0.05. Except for “pleasant dreams”, the odds ratio for anxiety disorder symptoms tended to increase with the frequency of the dream characteristic. For the following dream characteristics, participants in the “often” category had a higher likelihood of experiencing anxiety disorder symptoms than those in the “almost never” category: “rumination at bedtime” (aOR: 10.60; 95% CI: 5.92–18.97), “memory of dreams” (aOR: 2.86; 95% CI: 1.53–5.34), “emotional carryover” (aOR: 3.44; 95% CI: 1.87–6.32), “awakening by frightening dreams” (aOR: 3.53; 95% CI: 2.03–6.12), “unpleasant dreams” (aOR: 5.25; 95% CI: 2.86–9.64), “film-like dreams” (aOR: 2.18; 95% CI: 1.20–3.94), and “fantastical dreams” (aOR: 2.88; 95% CI: 1.62–5.13). For “pleasant dreams”, the odds ratio decreased gradually (aOR: 0.50; 95% CI: 0.25–0.97). Meanwhile, the odds ratio for “recurring dreams” was not statistically significant.

Similarly, the odds ratio for depressive symptoms tended to increase with the frequency of the dream characteristics, except for “pleasant dreams”. For the following dream characteristics, participants in the “often” category had a higher likelihood of experiencing depressive symptoms than those in the “almost never” category: “rumination at bedtime” (aOR: 8.79; 95% CI: 5.58–13.87), “memory of dreams” (aOR: 2.15; 95% CI: 1.40–3.30), “emotional carryover” (aOR: 4.43; 95% CI: 2.45–8.00), “awakening by frightening dreams” (aOR: 4.31; 95% CI: 2.66–6.99), “unpleasant dreams” (aOR: 10.13; 95% CI: 5.57–18.44), “film-like dreams” (aOR: 2.49; 95% CI: 1.60–3.86), “fantastical dreams” (aOR: 3.36; 95% CI: 2.19–5.18), and “recurring dreams” (aOR: 1.98; 95% CI: 1.18–3.31). For “pleasant dreams”, the odds ratio was lower in the “sometimes” category (aOR: 0.69; 95% CI: 0.48–0.99) compared with that in the “almost never” category.

## 3. Discussion

This study explored how the frequency of four dream-related experiences (A-1 to A-4) and five types of dream content (B-1 to B-5) were related to the presence of anxiety disorder symptoms and depressive symptoms among Japanese high school students. The odds ratios for both anxiety disorder and depressive symptoms tended to increase with the frequency of “rumination at bedtime”, “memory of dreams”, “emotional carryover”, “awakening by frightening dreams”, “unpleasant dreams”, “film-like dreams”, and “fantastical dreams”. In contrast, participants who often experienced “recurring dreams” exhibited a higher likelihood of only depressive symptoms (compared with those in the “almost never” category). Regarding “pleasant dreams”, the odds ratio decreased gradually for both anxiety disorder and depressive symptoms. Furthermore, “rumination at bedtime” and “unpleasant dreams” exhibited particularly high odds ratios, which indicated that they had a strong link to the deterioration of mental health among Japanese high school students.

The findings of this study aligned with those of previous studies that linked nightmares and emotional dreams with the deterioration of mental health [24,25]. Additionally, while previous studies have emphasized that dreams can serve as significant indicators of one’s mental and emotional state [20,23], this study introduces a new perspective by examining the impact of “rumination at bedtime” on one’s dreams. In particular, our results suggest a relationship between “rumination at bedtime” and mental health. Additionally, previous research revealed that “pleasant dreams” were more common in individuals with high extroversion, low neuroticism, and high openness [38]. Furthermore, this study reinforces the idea that “pleasant dreams” become less frequent as mental health deteriorates. Although the association between “unpleasant dreams” and mental health aligned with those of previous findings, “rumination at bedtime” may contribute to an increased stress load, which could lead to further long-term mental health deterioration. Notably, this study observed a tendency for the odds ratio to increase with an increase in the frequency of “rumination at bedtime”. This result suggested that “rumination at bedtime” may both reflect and contribute toward deteriorating mental health. The “continuity hypothesis” of dreaming [20,37] suggests a meaningful connection between waking life and dream content, with emotions, concerns, and even daily activities often being reflected in dreams. Stressful interpersonal events and negative emotional states during wakefulness may manifest as distressing or repetitive elements in dreams. This phenomenon has been observed in both clinical and non-clinical populations, which supports the idea that dreams can mirror ongoing psychological conflicts [37]. Moreover, research indicates that the extent to which waking experiences carry over into dreams may be influenced by individual traits, such as personality characteristics and stress sensitivity [20]. Empirical studies support this bidirectionality. The “continuity hypothesis” of dreaming suggests that waking emotional experiences, particularly those associated with stress, worry, or trauma, carry over into dreams [20,21]. Conversely, emotionally negative dream experiences may lead to sleep fragmentation, heightened emotional reactivity, and further stress upon waking, which may potentially exacerbate anxiety or depressive symptoms [24,28]. Hence, dream content reflects psychological states and also acts as a contributing factor to emotional dysregulation. Furthermore, considering the broader social and cultural context affecting Japanese adolescents, such as academic pressure and stigma surrounding mental health, is important. These factors may influence their psychological well-being and the emotional content and themes of their dreams [2,39,40].

Dreams are closely linked to brain activity during REM sleep [41], a period which is often associated with vivid, memorable dreams [29,30]. During REM sleep, part of the prefrontal cortex becomes suppressed, while the limbic system (including the amygdala, hippocampus, and cingulate gyrus), which is involved in emotions and memory, becomes activated [42,43]. Consequently, dream content tend to strongly reflect emotional states, and if a person is experiencing anxiety or depressive symptoms, this may manifest as nightmares. The “Activation–Synthesis hypothesis” [44] and its developmental version [45] are key theories explaining how dreams are formed. According to these hypotheses, the cerebral cortex integrates random electrical signals from the brainstem during REM sleep. It interprets them as dreams, and past memories and associations are incorporated to provide a structure to the dream. In individuals with anxiety or depression, overactivation of the limbic system could make their dream content more likely to reflect discomfort or fear. Another theory, the “sensory image-free association hypothesis”, [46] suggests that dreams are associative stories that begin with random visual images. Hence, if a person experiences anxiety or depression, they may engage in “rumination at bedtime”, wherein emotions related to past experiences, failures, and worries are replayed, which can potentially influence dream content and lead to nightmares. Rumination is a repetitive thought pattern focused on negative events, failures, and anxieties, and thus contributes to increased mental stress [47,48]. Such thought processes can affect brain activity during sleep, which can potentially intensify anxiety and depression symptoms and make them more likely to manifest as nightmares.

This study has several limitations. First, it was a cross-sectional study. Future longitudinal studies should verify causal relationships. Ideally, such studies could include a follow-up assessment 2–3 months later and incorporate the use of dream diaries. This approach would allow for further continuous data collection on dream characteristics and a clearer examination of temporal relationships with mental health symptoms. Second, our participants were from high schools located in a single region of Japan. This could have led to the possibility of selection bias. Since the frequency and characteristics of dreams can be affected by cultural factors and individual differences, a nationwide study is necessary. Third, previous studies have generally been conducted in Western countries, and while research has revealed that broad dream categories (e.g., “bad dreams”) are universal (i.e., culture-independent), cultural differences may affect the content of dreams (e.g., “what happens in a bad dream”) [49]. Fourth, since this study used a questionnaire to investigate dreams, comparing it with previous studies is difficult. Future research should use the Typical Dreams Questionnaire and Dream Characteristics Rating Scale (DP scale). Moreover, a detailed study of dreams must be conducted, including country-based comparisons. Fifth, individuals are unlikely to remember all the dreams they experience during the night. Nonetheless, to the best of our knowledge, this study is the first to comprehensively examine adolescent dreams and mental health in Japan. Its findings can serve as foundational data for future research.

## 4. Methods

### 4.1. Participants

Of the 19 high schools in a prefecture in western Japan, nine agreed to participate. Explanations were provided to all the participating schools, and approval was obtained from the principals with the cooperation of the nurses. This survey was conducted from April to June 2023 and involved 1060 high school students.

Students aged 15–18 years were included if both they and their parents/guardians provided informed consent to participate. Those who did not provide the required consent or submitted incomplete responses were excluded.

### 4.2. Survey Procedure

Data were collected via a survey. Before the survey was conducted, students from the participating schools received a written explanation of the study’s purpose and methodology. They were informed that participation was voluntary, their responses would not affect their grades, and the survey would be anonymous. Their parents received similar information in writing through the school. Those who did not wish for their child to participate were instructed to notify the school. This study was approved by the Ethics Committee of Nihon University School of Medicine (No. 2022-11). Informed consent was obtained before the survey form was administered.

### 4.3. Survey Questionnaire

We conducted the survey via an anonymous self-administered questionnaire that collected data on general characteristics (gender, grade, and age), lifestyle habits (breakfast frequency, exercise frequency, and weekday sleep duration), and dreaming frequency. It also included the Generalized Anxiety Disorder-7 (GAD-7), Patient Health Questionnaire modified for Adolescents (PHQ-A), and nine questions regarding dreams. Additionally, the Internet Addiction Test Diagnostic Questionnaire (DQ) [50] was included to assess internet addiction.

Previous studies suggested that lifestyle habits and problematic internet use were associated with both sleep quality and mental health symptoms. Inclusion of these variables allowed us to explore whether such factors influenced the relationship between dreams [6,16,50].

### 4.4. Measures

#### 4.4.1. Mental Health and Lifestyle Habits

Symptoms of anxiety disorder and depressive symptoms were measured via the GAD-7 and PHQ-A, respectively. Both the GAD-7, a scale used to measure generalized anxiety disorder [51], and PHQ-A, which assessed both physical and mental health conditions and included depressive symptoms [52], were confirmed as reliable and valid for adolescents. The PHQ-A is a revised version of the Patient Health Questionnaire-9. It was designed to measure the severity of depressive symptoms in adolescents and was valid [53]. Japanese versions of both these scales demonstrated good reliability and validity. Previous studies revealed high internal consistency and confirmed their construct validity in Japanese clinical and general populations [54].

Results for the PHQ-A and GAD-7 were examined based on the cut-off points for the Japanese version. A score of ≤4 and ≥5 points on the PHQ-A denoted the absence and presence of depressive symptoms, respectively. Similarly, a score of ≤10 and ≥11 points on the GAD-7 denoted the absence and presence of anxiety disorder symptoms, respectively.

Regarding lifestyle habits, breakfast frequency was categorized as “every day”, “occasionally”, and “rarely”. Exercise frequency was categorized as “≥4 times a week”, “1–4 times a week”, and “never”. Weekday sleep duration was categorized as “>8 h”, “6–8 h”, and “<6 h”.

#### 4.4.2. Dream-Related Questions

Dream-related questions asked participants regarding the presence or absence of certain dream-related experiences and dream content in the last 30 days. Section A comprised four questions on dream-related experiences. Section B comprised five questions on dream content. Responses included “almost never”, “sometimes (sometimes have dreams/experiences of that type)”, or “often (often have dreams/experiences of that type)”. The content of each section is given below.

Section A. Questions about dream-related experiences in the last 30 days:
Did you experience unpleasant thoughts before going to sleep? (assessed “rumination at bedtime”)Do you remember most of the dreams you had? (assessed “memory of dreams”)Did your dreams affect your mood the next day? (assessed “emotional carryover”)Were you woken up by frightening dreams? (assessed “awakening by frightening dreams”)
Section B. Questions about dream content in the last 30 days:
Did you have any pleasant dreams? (assessed “pleasant dreams”)Did you have any unpleasant dreams? (assessed “unpleasant dreams”)Did you have any dreams in which you could see clear, movie-like images? (assessed “film-like dreams”)Did you have any strange dreams that could not be real? (assessed “fantastical dreams”)Did you have the same dream over and over again? (assessed “recurring dreams”)


Questions A-4 (awakening by frightening dreams) and B-2 (unpleasant dreams) were included based on the relationship between nightmares and mental health observed in previous studies [21,22,23,24,25,37]. Other questions addressed different dream-related experiences and dream content.

Pearson’s product–moment correlation coefficients and *p*-values were calculated to explore the relationships between various dream-related experiences and dream content, which included the correlations between different dream-related experiences, as well as distinct types of dream content, as summarized in Table 1. The relationship between different dream characteristics ranged from weak to moderate. Statistically significant correlations were found for most combinations, except between “pleasant dreams” and “rumination at bedtime”, and “fantastical dreams” and “recurring dreams”. These results suggested a meaningful correlation from a categorical perspective.

### 4.5. Data Analysis

First, we determined the descriptive statistics (frequencies and percentages) for participants’ characteristics, as well as prevalence of anxiety disorder and depressive symptoms based on these characteristics. Chi-squared tests were performed to identify significant differences in the prevalence of anxiety disorder and depressive symptoms based on participants’ characteristics.

Second, we determined the descriptive statistics (frequencies and percentages) for dream-related experiences and dream content, as well as prevalence of anxiety disorder symptoms and depressive symptoms based on the frequency and content. Additionally, we used *p*-trends to assess whether the prevalence of anxiety disorder and depressive symptoms changes based on the frequency of dream-related experiences and dream content.

Furthermore, we performed logistic regression analysis to examine the association between the frequency of each dream-related experience and dream content and presence of anxiety disorder and depressive symptoms. Gender, grade, breakfast frequency, exercise frequency, internet addiction, weekday sleep duration, and dreaming frequency were set as covariates. This was as dream-related experiences and dream content were correlated (Table 5), which raised concerns regarding multicollinearity. We set the presence of anxiety disorder symptoms and presence of depressive symptoms as the dependent variables. Frequency of each dream-related experience and dream content was set as an independent variable. All statistical analyses were performed via STATA version 17.0, a statistical analysis software developed by StataCorp.

## 5. Conclusions

This study investigated the relationship between dream content and mental health among Japanese adolescents. The results revealed that higher frequencies of anxiety and depressive symptoms were associated with specific types of dreams, particularly rumination at bedtime and unpleasant dreams. These findings suggest a possible bidirectional relationship, in which mental health shapes dream content and, conversely, dream characteristics may influence psychological well-being. Notably, dream features could serve as early indicators of mental health deterioration.

Since adolescent mental health is an increasing concern, simple and effective methods for early detection are urgently required. Asking about specific dream experiences may offer a quick and accessible tool for screening. Since certain types of dreams, such as nightmares, can be modified through psychological interventions, such as imagery rehearsal therapy (IRT) [55]. Furthermore, integrating dream-related questions into school-based or home-based support programs may contribute to prevention and early intervention strategies both in Japan and internationally.

## Figures and Tables

**Table 1 clockssleep-07-00034-t001:** Participants’ general characteristics.

	Male	Female	Total
10th graders (15.05 ± 0.23 years old)	329	245	574
11th graders (16.03 ± 0.22 years old)	123	135	258
12th graders (17.05 ± 0.21 years old)	14	8	22
Total	466	388	854

**Table 2 clockssleep-07-00034-t002:** Participants’ characteristics and prevalence of anxiety disorder and depressive symptoms.

			Prevalence of Anxiety Disorder Symptoms		Prevalence of Depressive Symptoms	
	N	%	%	*p*-Value	%	*p*-Value
Gender						
Male	466	54.6	9.8	<0.001	32.1	<0.001
Female	388	45.4	17.1		43.7	
Grade						
10th grade	574	67.2	13.4	0.930	34.3	<0.001
11th grade	258	30.2	12.5		42.3	
12th grade	22	2.6	13.6		59.1	
Dreaming frequency						
Almost never	113	13.5	10.6	<0.001	24.8	<0.001
Sometimes	339	40.5	9.4		31.5	
Often	385	46.0	17.4		45.8	
Breakfast frequency						
Every day	706	82.9	11.9	<0.001	33.9	<0.001
Occasionally	79	9.3	13.9		48.1	
Rarely	67	7.9	25.4		64.1	
Exercise frequency						
Less frequently	74	8.7	16.2	0.240	37.0	0.120
Four or fewer times a week	451	52.8	14.4		40.4	
Four or more times a week	329	38.5	10.7		33.2	
Internet addiction						
Not addicted	758	89.0	10.9	<0.001	32.9	<0.001
Addicted	94	11.0	30.9		73.1	
Weekday sleep duration						
6 h or less	260	30.6	16.2	<0.001	48.5	<0.001
6–8 h	535	63.0	11.7		32.4	
8 h or more	54	6.4	11.5		33.3	
Anxiety disorder symptoms						
Absence of anxiety disorder symptoms	735	86.9	-	-	29.5	<0.001
Presence of anxiety disorder symptoms	111	13.1	-		87.4	
Depressive symptoms						
Absence of depressive symptoms	533	62.6	2.6	<0.001	-	-
Presence of depressive symptoms	318	37.4	31.0		-	

Note. Participants with missing values were excluded.

**Table 3 clockssleep-07-00034-t003:** Frequency of each dream-related experience and dream content and prevalence of anxiety disorder and depressive symptoms based on their frequency and content.

			Prevalence of Anxiety Disorder Symptoms		Prevalence of Depressive Symptoms	
	N	%	%	*p*-Trend	%	*p*-Trend
Rumination at bedtime (A-1)						
Almost never	455	54.0	4.4	<0.001	21.0	<0.001
Sometimes	222	26.3	13.1		43.9	
Often	166	19.7	37.3		72.9	
Memory of dreams (A-2)						
Almost never	284	33.7	7.7	<0.001	28.4	<0.001
Sometimes	328	38.9	13.4		34.9	
Often	231	27.4	19.5		51.5	
Emotional carryover (A-3)						
Almost never	667	79.2	10.0	<0.001	30.9	<0.001
Sometimes	105	12.5	20.0		55.2	
Often	70	8.3	32.9		71.4	
Awakening by frightening dreams (A-4)						
Almost never	519	61.6	8.7	<0.001	27.7	<0.001
Sometimes	212	25.1	15.6		44.8	
Often	112	13.3	29.5		67.0	
Pleasant dreams (B-1)						
Almost never	323	38.0	15.7	0.111	39.3	0.489
Sometimes	390	45.8	11.9		35.9	
Often	138	16.2	10.9		37.0	
Unpleasant dreams (B-2)						
Almost never	477	56.2	7.8	<0.001	21.2	<0.001
Sometimes	281	33.1	14.4		52.7	
Often	91	10.7	37.4		74.7	
Film-like dreams (B-3)						
Almost never	350	41.3	8.7	<0.001	27.7	<0.001
Sometimes	311	36.7	13.5		37.0	
Often	187	22.1	21.2		56.1	
Fantastical dreams (B-4)						
Almost never	354	41.7	8.2	<0.001	24.3	<0.001
Sometimes	302	35.6	12.0		38.7	
Often	193	22.7	24.2		59.1	
Recurring dreams (B-5)						
Almost never	599	70.6	11.8	0.020	32.4	<0.001
Sometimes	158	18.6	14.0		44.9	
Often	92	10.8	21.1		57.6	

Note. Participants with missing values were excluded.

**Table 4 clockssleep-07-00034-t004:** Association between frequency of each dream characteristic and presence of anxiety disorder and depressive symptoms.

		Presence of Anxiety Symptoms			Presence of Depressive Symptoms		
		aOR	95% CI	*p*-Value	aOR	95% CI	*p*-Value
Rumination at bedtime (A-1)	Almost never	1.00			1.00		
	Sometimes	3.10	1.67–5.74	<0.001	3.09	2.10–4.55	<0.001
	Often	10.60	5.92–18.97	<0.001	8.79	5.58–13.87	<0.001
Memory of dreams (A-2)	Almost never	1.00			1.00		
	Sometimes	2.11	1.16–3.85	0.015	1.16	0.78–1.72	0.452
	Often	2.86	1.53–5.34	0.001	2.15	1.40–3.30	<0.001
Emotional carryover (A-3)	Almost never	1.00			1.00		
	Sometimes	1.86	1.06–3.28	0.032	2.16	1.37–3.40	0.001
	Often	3.44	1.87–6.32	<0.001	4.43	2.45–8.00	<0.001
Awakening by frightening dreams (A-4)	Almost never	1.00			1.00		
	Sometimes	1.84	1.11–3.05	0.018	2.16	1.49–3.12	<0.001
	Often	3.53	2.03–6.12	<0.001	4.31	2.66–6.99	<0.001
Pleasant dreams (B-1)	Almost never	1.00			1.00		
	Sometimes	0.64	0.40–1.04	0.071	0.69	0.48–0.99	0.046
	Often	0.50	0.25–0.97	0.042	0.69	0.43–1.12	0.136
Unpleasant dreams (B-2)	Almost never	1.00			1.00		
	Sometimes	1.84	1.10–3.08	0.020	3.99	2.78–5.74	<0.001
	Often	5.25	2.86–9.64	<0.001	10.13	5.57–18.44	<0.001
Film-like dreams (B-3)	Almost never	1.00			1.00		
	Sometimes	1.70	0.98–2.93	0.059	1.46	1.00–2.13	0.047
	Often	2.18	1.20–3.94	0.010	2.49	1.60–3.86	<0.001
Fantastical dreams (B-4)	Almost never	1.00			1.00		
	Sometimes	1.42	0.80–2.51	0.229	1.78	1.21–2.61	0.003
	Often	2.88	1.62–5.13	<0.001	3.36	2.19–5.18	<0.001
Recurring dreams (B-5)	Almost never	1.00			1.00		
	Sometimes	1.02	0.59–1.77	0.932	1.32	0.89–1.97	0.170
	Often	1.35	0.72–2.53	0.342	1.98	1.18–3.31	0.009

Notes Adjustments were made for gender, grade, breakfast frequency, exercise frequency, internet addiction, weekday sleep duration, and dreaming frequency. Participants with missing values were excluded. aOR: adjusted odds ratio; CI: confidence interval.

**Table 5 clockssleep-07-00034-t005:** Correlations between dream-related experiences and dream content for model building.

	A-1	A-2	A-3	A-4	B-1	B-2	B-3	B-4	B-5
Rumination at bedtime (A-1)	-								
Memory of dreams (A-2)	0.248 **								
Emotional carryover (A-3)	0.300 **	0.374 **							
Awakening by frightening dreams (A-4)	0.271 **	0.358 **	0.385 **						
Pleasant dreams (B-1)	0.047	0.311 **	0.111 *	0.077 *					
Unpleasant dreams (B-2)	0.404 **	0.279 **	0.290 **	0.490 **	0.081 *				
Film-like dreams (B-3)	0.278 **	0.503 **	0.294 **	0.327 **	0.347 **	0.398 **			
Fantastical dreams (B-4)	0.288 **	0.329 **	0.240 **	0.311 **	0.262 **	0.392 **	0.462 **		
Recurring dreams (B-5)	0.239 **	0.316 **	0.288 **	0.311 **	0.271 **	0.352 **	0.426 **	0.309	-

Note. * *p*-value < 0.050, ** *p*-value < 0.001.

## Data Availability

The data that support the findings of this study are available from the corresponding author, YO, upon reasonable request.

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
