# Peer review of "Association Between Dreams, Anxiety, and Depressive Symptoms Among Japanese Adolescents: A Cross-Sectional Study"

_2624-5175, 2025, doi:10.3390/clockssleep7030034_

Round 1
Reviewer 1 Report
Comments and Suggestions for Authors
Youth depression and suicide are major problems in developed countries with declining birthrates, especially in East Asia. This paper, which evaluates the association of dream content as a screening or predicting of depression, is significant. On the other hand, it is already well known that dream content reflects psychiatric symptoms, and the novelty of this study is somewhat limited. To compensate for this shortcoming, it would be better to deepen the discussion of the bidirectional relationship between dreams and mental states in the “Discussion” section. It might also be better to supplement this section with a discussion of the social environment and cultural influences on Japanese youth.
Author Response
Dear Reviewer,
We would like to sincerely thank the reviewer for their thoughtful and constructive feedback. We are grateful that you recognized the significance of evaluating the association between dream content and mental health among adolescents, particularly in East Asia where youth depression and suicide are critical social concerns.
We also appreciate your valuable suggestions regarding the need to strengthen the discussion of the bidirectional relationship between dreams and mental states, as well as influence of social and cultural factors on Japanese youth.
Comment. Youth depression and suicide are major problems in developed countries with declining birthrates, especially in East Asia. This paper, which evaluates the association of dream content as a screening or predicting of depression, is significant. On the other hand, it is already well known that dream content reflects psychiatric symptoms, and the novelty of this study is somewhat limited. To compensate for this shortcoming, it would be better to deepen the discussion of the bidirectional relationship between dreams and mental states in the “Discussion” section. It might also be better to supplement this section with a discussion of the social environment and cultural influences on Japanese youth.
Response: Thank you for your comment. We have revised the Discussion section accordingly (marked p. 7, lines 20-39).
Furthermore, we have incorporated a detailed explanation of the continuity hypothesis of dreaming, supported by relevant empirical studies [now cited as references 20 and 38], to frame the dynamic interaction between waking emotional states and dream content. We have also discussed how both clinical and non-clinical populations may exhibit this pattern.
We have also expanded on the bidirectional nature of the relationship between negative dream experiences and mental health symptoms, and emphasized how certain dream characteristics may reflect and contribute to psychological distress.
Furthermore, we have added a paragraph discussing sociocultural influences relevant to Japanese adolescents, such as academic pressure and mental health stigma. These contextual factors may affect both their mental well-being and the emotional tone of their dreams.
We hope these revisions successfully address your concerns and enhance our manuscript’s scholarly contribution.
Thank you again for your thoughtful review and for helping us improve the quality and clarity of our work.
Reviewer 2 Report
Comments and Suggestions for Authors
Please see as attached

Author Response
Dear Reviewer,
We sincerely thank you for your thoughtful and detailed review of our manuscript. We appreciate your encouraging comment on recognizing the value of our study on the association between dream characteristics and mental health among Japanese adolescents.
We have carefully considered your comments and revised the manuscript accordingly. Please find our point-by-point responses below:
Comment 1. Authors have mentioned the aim of the study in the Introduction section. Readers would like to know what the objectives of the study are - stating primary, and secondary objectives if any.
Response:Thank you for your comment. We have revised this section to clearly state the primary objectives of this study (marked p. 2, lines 51-53 and p.3 lines1-2).
Comment 2 and 3. Method. Inclusion and exclusion criteria are not stated. Please describe inclusion and exclusion criteria.
Response:Thank you for your feedback. We have revised the Methods section to include detailed inclusion and exclusion criteria for participant selection. (marked p. 8, lines 34-36).
Comment 4. It is not clear how lifestyle habits and internet addiction are related to the aims and objectives of this study. Any such presumed relationship has not been addressed in the introduction/background section.
Response:We have revised the Survey Questionnaire subsection to explain the rationale for including lifestyle habits and internet addiction in accordance with your comment. Furthermore, we have citied relevant literature that support their potential association with adolescent mental health and dream content. (marked p.9, lines 6-8).
Comment 5. Can the authors let us know the psychometric properties of the Japanese versions of the PHQA and GAD-7?
Response:Thank you for your valuable comment. We have added additional information on the validity and reliability of the Japanese versions of the PHQ-A and GAD-7, with references to validation studies in adolescent populations (marked p. 9, lines 17-19).
Comment 6. Table 1 needs to be properly formatted - g dreams (A-4) seem out of place.
Response:Thank you for your observation. We have carefully reviewed and adjusted the formatting of revised Table 1 for additional clarity. The Method section has been moved, so Table 1 has been changed to Table 5. Specifically, we modified the column headers to reveal only item numbers (e.g., A-1 to B-5), while retaining the full variable names in the row labels, thus, improving readability and preventing confusion. (marked p. 10 Table 5).
Comment 7. Table 1 appears under Methods section. Should it be in Results section?
Response:We understand your concern. However, we believe Table 1 is best placed in the Methods section, as it provides correlation data used to inform our modeling strategy rather than to present the results. The Method section has been moved, so Table 1 has been changed to Table 5.
To clarify its role, we have revised the title to include the phrase “for model building” to indicate that it is part of the analytic planning process (marked p. 10 Table 5).
Comment 8. In discussing the limitations of the study the authors have one line – future longitudinal studies should verify causal relationship. Can the authors expand on how this could be done?
Response:We have expanded the limitations to describe how future longitudinal designs, such as follow-up assessments and dream diaries, could help determine causality between dream features and mental health outcomes. (marked p. 8, lines 10-13).
Comment 9. The Conclusion is too long. Ideally one paragraph long or two or three brief paragraphs that effectively summarize the main points of the manuscript. Authors should consider making their conclusion concise.
Response:We have shortened and restructured the conclusion to summarize the key findings and implications in two concise paragraphs in line with your recommendation. (marked p. 11, lines 5-18).
Comment 10. Patents (page 14): not clear what this means in the context of this manuscript. A patent means and official document that confers a right or privilege. Perhaps this is a literal translation of some word. Authors should consider deleting it, or explain how it fits in the manuscript.
Response:We thank the reviewer for pointing this out. We agree that the heading “Patents” was inappropriate and potentially misleading as the manuscript does not contain any patent-related content. Therefore, we have deleted this section heading.
We hope that these revisions sufficiently address the reviewer’s comments. Thank you for your time and helpful feedback, which significantly improved the quality of our manuscript.